# AUTH-PROMPT BENCH: TOWARDS RELIABLE AND STABLE PROMPTING IN TEXT-TO-IMAGE GENERATION

## ABSTRACT

Recent advances in diffusion models and large language models (LLMs) have enabled text-to-image generation of remarkable fidelity, allowing users to synthesize images directly from natural language prompts. However, we observe that prompt informativeness varies significantly across user proficiency levels, which draws little research attention in the field. Those ambiguous or under-specified prompts often lead to unstable outputs that deviate from user intent, while current benchmarks provide limited means to quantify this phenomenon. We address this gap by introducing Authentic Prompt Benchmark (Auth-Prompt Bench), a large-scale benchmark of 17,580 prompt-image pairs from both novice and expert users sourced from authentic web cases, specifically designed to evaluate the stability of the prompting in text-to-image generation. Unlike existing metrics that focus solely on prompt–image alignment, Auth-Prompt Bench is further grounded in an information-theoretic perspective of prompt-to-prompt transmission, enabling stability assessment through complementary metrics: mutual information, prompt entropy, and prompt energy. Building on these insights, we propose NoxEye, an end-to-end prompt optimization framework comprising (i) an information enhancer that maps user prompts toward the model-preferred distribution, and (ii) an information aligner that enforces fine-grained alignment of visual entities. Across Auth-Prompt Bench and other established benchmarks, NoxEye delivers improvements of up to 13.52% in mutual information, 20.30% in prompt entropy, and 27.01% in prompt energy and enhances prompts from novice users, over state-of-the-art baselines. Our results establish Auth-Prompt Bench as one of the first dedicated benchmarks for stability in T2I generation and demonstrate that information-theoretic prompt optimization can significantly enhance both robustness and fidelity. The human evaluation results further verify the efficacy of our method in user intent alignment for T2I generation. We hope this work provides a foundation for the community on principled evaluation and reliable user–model interaction in T2I generative systems. The source code and dataset will be made publicly available.

## 1 INTRODUCTION

With the advent of diffusion models (Rombach et al., 2022; Ho et al., 2020; Ramesh et al., 2021; 2022; Saharia et al., 2022; Jiang et al., 2024), text-to-image generation has become increasingly popular, enabling users to generate images based on a wide variety of textual prompts. The development of large language models (LLMs) has further enhanced this process by allowing for prompt tuning, leading to improved visual fidelity in the generated images(Hao et al., 2023; Wu et al., 2024; Yang et al., 2024).

However, due to the significant gap in user proficiency levels, some vague or ambiguous prompts from novice users may fail to produce images aligned with the user's intentions. Some pioneering research (Du et al., 2023; Chefer et al., 2023b) has demonstrated that diffusion models exhibit instability when interpreting such prompts. Notably, even minor lexical noise in a prompt can confuse the text encoder, producing images that fail to capture the intended semantics.

Our rationale stems from the observation that prior work (Hao et al., 2023; Cao et al., 2023; Rosenman et al., 2023) mainly addresses text–image alignment or aesthetic quality, but pays little attention to **user intent** and **prompt informativeness**, leaving generation stability insufficiently explored.

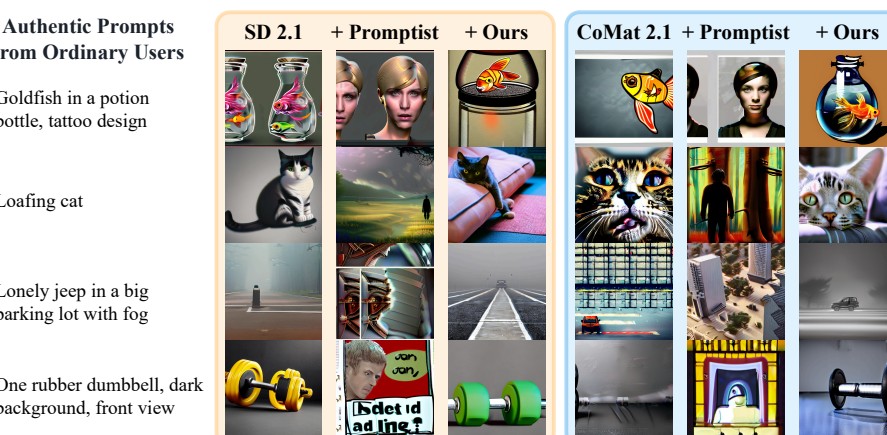

Figure 1: We observe that state-of-the-art diffusion models, such as Stable Diffusion 2.1 (Rombach et al., 2022) and CoMat 2.1 (Jiang et al., 2024), remain prone to generation instability under authentic user inputs. Our prompt-optimizing method, NoxEye, mitigates this limitation by systematically enhancing prompt–image alignment with user intent and outperforms the state-of-the-art prompt-refining method, e.g., Promptist (Hao et al., 2023).

This gap is especially problematic given the heterogeneity of user proficiency: while expert users often craft long, detailed prompts with explicit attributes, novice users tend to produce shorter and more ambiguous descriptions. Consequently, we argue that advancing the stability and reliability of diffusion-based T2I systems requires both the design of interpretable prompts and the development of robust evaluation metrics.

We cast the T2I prompt stabilizing problem as a distribution matching problem, where the goal is to train a model such that the conditional distribution of the optimized prompt given user authentic prompts better resembles user intents. To this end, we propose **NoxEye**, a plug-and-play modular prompt optimization framework broadly compatible with diverse T2I diffusion models. NoxEye consists of two main components: (i) an *information enhancer*, which maps prompts from the human-interpretable distribution to the model-preferred distribution, and (ii) an *information aligner*, which further guides the enhanced prompts to achieve fine-grained visual entity alignment.

To facilitate the study for the community, we additionally release a novel benchmark, named Auth-Prompt Bench, for assessing the stability of T2I generation through the lens of information propagation. Unlike prior efforts, our benchmark not only measures the alignment between generated images and user intent but also explicitly quantifies the information content embedded in prompts. To support this, we curate a dataset of 17,580 real-world prompts crawled from authentic web cases, stratified into *novice* and *expert* subsets according to user proficiency in T2I prompt design.

Extensive experiments demonstrate that NoxEye effectively mitigates the adverse effects of ambiguous prompts, substantially improving both fidelity and stability of generated images. When applied to existing advanced T2I models, our framework yields 13.52%, 7.9% and 35.69% gains in mutual information, prompt entropy, and prompt energy metrics. Moreover, compared to the state-of-the-art prompt optimization strategies, NoxEye achieves a 23.20% and 8.34% improvement in mutual information and prompt energy metrics, underscoring its superior robustness to vague or underspecified text inputs.

## 2 RELATED WORK

### 2.1 TEXT-TO-IMAGE GENERATION AND BENCHMARKS

Text-to-image generation has progressed from early GANs (Goodfellow et al., 2014) and VAEs (Kingma & Welling, 2013) to diffusion-based models, with Stable Diffusion (Rombach et al., 2022) and CoMat (Jiang et al., 2024) exemplifying the current paradigm. These models typically employ frozen text encoders (e.g., CLIP (Radford et al., 2021)) to map prompts to embeddings that guide iterative denoising.

Benchmarking efforts have evolved alongside model capabilities. HEIM (Liang et al., 2022) evaluates twelve dimensions, including alignment, quality, reasoning, and fairness. T2I-CompBench (Huang et al., 2025) focuses on compositional generation with novel metrics and reward-driven fine-tuning (GORS), while GenEval (Ghosh et al., 2023) introduces object-centric evaluation for fine-grained analysis. Despite these advances, models still struggle to capture user intent accurately, motivating the proposed Authentic Prompt Benchmark for mapping ambiguous prompts to concrete object representations.

## 2.2 PROMPT OPTIMIZATION FOR TEXT-TO-IMAGE GENERATION

Prompt optimization leverages LLMs (Schlegel et al., 2025; Xiang et al., 2025) to improve generated image quality. Promptist (Hao et al., 2023) fine-tunes GPT-2 (Radford et al., 2019) to reformulate user prompts via supervised fine-tuning (SFT) and direct preference optimization (DPO) using CLIP similarity and aesthetics scores. Self-Rewarding LVLMs (Yang et al., 2025) extend this two-stage paradigm with a self-reward mechanism, while PAG (Yun et al., 2025) uses GFlowNets to generate diverse adaptive prompts.

These approaches enhance prompt quality but largely focus on aesthetic and relevance objectives, often neglecting whether generated images faithfully reflect the user's underlying intent.

## 3 OUR ROADMAP TO AUTH-PROMPT BENCH

### 3.1 UNDERLYING RATIONALE

Our rationale stems from a key observation: due to user proficiency level, the informativeness of prompts exerts a substantial impact on the quality of text-to-image (T2I) generation, while the community has paid limited attention to systematically addressing this issue. Motivated by this gap, our objective is to design a comprehensive and principled methodology and benchmark for evaluating the quality of prompts in T2I generation.

Below, we demonstrate how we quantify the informativeness of a text-to-image prompt $P$ in conveying user intent $Y$ to a generative model producing image $I$, using three carefully designed measures from an information-theoretic framework.

#### 3.1.1 MUTUAL INFORMATION FOR USER INTENT ALIGNMENT

We model generation as a Markov chain $Y \to P \to I$, and define prompt stability via mutual information:

$$I(Y; I) = H(Y) - H(Y \mid I),$$

where larger $I(Y; I)$ indicates better preservation of user intent.

Since $p(y \mid I)$ is intractable, we approximate it with a CLIP-based (Radford et al., 2021) predictive distribution $q(y \mid I)$, yielding

$$I(Y; I) \approx H(Y) + \mathbb{E}_{I,Y} \log q(Y \mid I).$$

Following prior work (Du et al., 2023; Feng et al., 2022; Chefer et al., 2023a), user intent is approximated via **entity–template expansion**, $Y(e) = \{t_k(e)\}_{k=1}^K$, allowing stability assessment over a distribution of plausible prompts. CLIP-based (Radford et al., 2021) similarity defines

$$q(y \mid I) \propto \exp\left(\tau\, s(I, y)\right), \quad s(I, y) = \frac{\phi_{\text{img}}(I) \cdot \phi_{\text{text}}(y)}{\|\phi_{\text{img}}(I)\| \|\phi_{\text{text}}(y)\|}.$$

Fano's inequality (Verdú et al., 1994) links mutual information to classification error $e$:

$$I(Y; I) \geq \log K - H(e) - e \log(K - 1),$$

which directly establishes the connection between intent-recovery accuracy and prompt stability. The inequality arises after a sequence of derivations (see Appendix B.1 and B.2).

Table 1: Example structure of Auth-Prompt Bench. Prompts are sourced from raw, real-world web cases, ensuring authenticity and diversity of user prompting.

| Intent category | Example (novice) | Example (expert) |
|---|---|---|
| Great white shark | Shark with a creepy smile in a cartoon style | Great White Shark poorly attempting to disguise itself as a Catholic Priest, 8K, 4K, HDRI |
| Hen | A chicken | Hen, village, overcast, glow, red |
| Television | A hyper realistic 2000s tv | Television set dressed like a brutal eagle, analog photo, 1850s london, 35mm, street scene |

### 3.1.2 PROMPT ENTROPY FOR T2I RELIABILITY ASSESSMENT

To quantify the informativeness of user inputs (Farquhar et al., 2024; Cheng et al., 2025; Duan et al., 2023), we introduce the notion of **prompt entropy**. Intuitively, novice users often provide under-specified or ambiguous prompts that lack sufficient detail, making them harder to interpret and yielding unstable generations. In contrast, expert prompts tend to be more specific and constrained, thereby concentrating information and reducing uncertainty.

Thus, we introduce the T2I Prompt entropy $H(P)$ reflects the inherent information content of $P$:

$$H(P) \approx -\frac{1}{T} \sum_{t=1}^{T} \log p_\theta(w_t \mid w_{<t}),$$

where $p_\theta$ is a pretrained LM. Lower entropy prompts are more predictable, concentrate information, and typically yield more stable generations. See Appendix B.3 for derivation and theoretical connection to $I(Y; I)$.

### 3.1.3 PROMPT ENERGY FOR T2I STABILITY ASSESSMENT

Existing stability metrics for text-to-image generation, such as classification accuracy or prompt entropy, capture either end-to-end information transfer or prompt *aleatoric uncertainty*, but fail to capture the model's *epistemic uncertainty*—uncertainty stemming from the model's lack of knowledge (Ma et al., 2025). To address this, we introduce **prompt energy** as a complementary measure: prompts with low energy correspond to concepts well-represented in the model, yielding stable generation, while high-energy prompts indicate unfamiliar or uncertain concepts. Formally, for a prompt sequence $x = (x_1, \ldots, x_T)$, the normalized sequence energy is

$$E(x) = -\frac{1}{T} \sum_{t=1}^{T} z_t(x_t),$$

where $z_t(x_t)$ denotes the model-assigned logit for token $x_t$. Lower $E(x)$ indicates higher confidence, whereas higher $E(x)$ signals uncertainty.

Combining prompt-level entropy and energy with end-to-end metrics such as $I(Y; I)$ provides a more comprehensive characterization of generation stability, directly linking user-provided information to image fidelity. Implementation details, derivations from model logits, and the connection to classical Boltzmann energy are provided in Appendix B.4.

## 3.2 BENCHMARK CONSTRUCTION

Building on our information-theoretic formulation, we design a benchmark to empirically evaluate prompt informativeness and generation stability. Inspired by ImageNet (Russakovsky et al., 2015), we curate 1,000 carefully selected intent categories, each paired with a set of text-to-image prompts and their corresponding outputs.

To capture variability in user expertise, prompts are carefully stratified into two groups: *novice* and *expert*. Novice prompts, sourced from `Lexica` (https://lexica.art/), reflect typical users who provide shorter, less informative descriptions. Expert prompts, collected from `Civitai` (https://civitai.com/), often specify detailed attributes, yielding richer, higher-information prompts. The two types of prompts are **manually re-verified** to ensure that: (1) novice and expert prompts strictly adhere to their intended styles, and (2) the corresponding images are filtered to

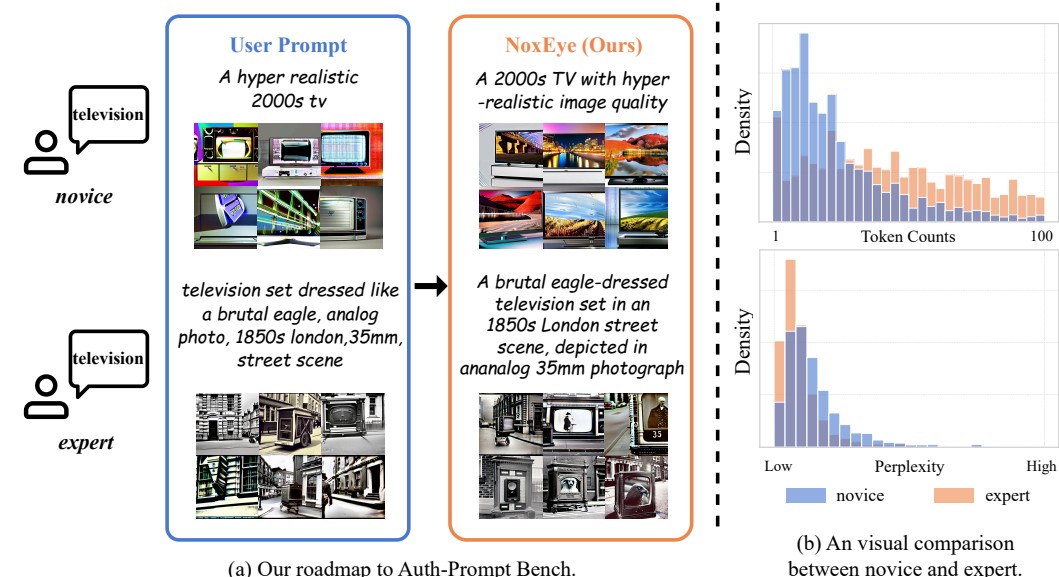

(a) Our roadmap to Auth-Prompt Bench.

(b) An visual comparison between novice and expert.

Figure 2: Instability of real-world prompts and importance of Auth-Prompt Bench. We curate a dataset of 17,580 real-world prompts collected from authentic web sources, stratified into novice and expert subsets, and show that novice prompts exhibit more tokens and higher perplexity compared to expert prompts. As shown in (a), the prompt from the novice user leads to unstable T2I outcomes, and our method can mitigate the issue by aligning the prompt to a model-friendly distribution.

guarantee ethical compliance, safety, and the absence of sensitive content. Each intent category contains up to 10 instances, with each instance comprising (i) the user prompt, (ii) generated image URL, (iii) generation parameters, and (iv) auxiliary metadata.

This benchmark enables systematic evaluation of how prompt informativeness—quantified via mutual information, prompt entropy, and prompt energy—affects generation stability. Mutual information measures the end-to-end alignment between user intent and generated images, reflecting whether the prompt provides sufficient information for semantically correct outputs. Prompt entropy, estimated using language model cross-entropy, captures the descriptive richness of the prompt. Prompt energy evaluates the model's internal "trust" in the input by measuring compatibility with its learned representation space.

Together, these three metrics form a triangulated evaluation framework: mutual information provides an empirical upper bound of stability, prompt entropy assesses intrinsic informativeness, and prompt energy gauges the model's internal calibration. Metrics are aggregated at both instance and class levels. Mutual information is evaluated via classification accuracy using a CLIP (Radford et al., 2021) zero-shot classifier, while prompt entropy and energy are measured with Gemma-3-1B (Gemma Team, 2025), offering fine-grained insights into the impact of prompt characteristics on generation stability across user expertise levels.

## 4 NOXEYE: AN END-TO-END PROMPT OPTIMIZATION FRAMEWORK

We aim to improve the stability of text-to-image generation by aligning user-provided prompts with the preference subspace of the target generative model. The generation process is formalized in Section 3.1. According to the **data processing inequality**, the end-to-end stability—quantified by the mutual information $I(Y; I)$ between the generated image $I$ and the intended output $Y$—is upper-bounded by the information encoded in the input prompt $P$. Therefore, to increase $I(Y; I)$, our rationale is to map user prompts closer to the **model's preferred prompt subspace**, with the prompt of which the model is most responsive and stable.

Figure 3: Overview of NoxEye training and inference.

## 4.1 PREFERENCE DISTRIBUTION AND OBJECTIVE FORMULATION

To formalize this idea, we introduce an **ideal preference distribution** $\mathcal{D}_{\text{pref}}$, which represents the set of prompts that are known to consistently yield high-fidelity and stable images under the target generative model. Given a user prompt $P_{\text{user}}$, our goal is to learn a transformation

$$P_{\text{stable}} = f_\theta(P_{\text{user}}),$$

parameterized by $\theta$, such that the resulting distribution $p_\theta(P^*)$ aligns with $\mathcal{D}_{\text{pref}}$. Using a distance measure $d(\cdot, \cdot)$, the optimization objective can be written as

$$\theta^* = \arg\min_\theta \ \mathbb{E}_{(P_{\text{user}}, I)} \Big[ d\big( f_\theta(P_{\text{user}}), \mathcal{D}_{\text{pref}} \big) \Big],$$

where $(P_{\text{user}}, I)$ denotes a training pair consisting of the user prompt and its corresponding generated image. In practice, we instantiate the distance as **Kullback–Leibler divergence**, leading to the following optimization problem:

$$\mathcal{L}_{\text{KL}}(\theta) = D_{\text{KL}}\big(p_\theta(P^*) \,\|\, \mathcal{D}_{\text{pref}}\big).$$

## 4.2 PREFERENCE DISTRIBUTION EXTRACTOR

The preference distribution $\mathcal{D}_{\text{pref}}$ is not directly available and must be estimated. To this end, we design a **preference extractor** $g_{\text{pref}}$ that generates a high-fidelity textual description $P_{\text{pref}}$ from an image $I$:

$$P_{\text{pref}} = g_{\text{pref}}(I), \quad \mathcal{D}_{\text{pref}} \approx p(P_{\text{pref}} \mid I).$$

In practice, $g_{\text{pref}}$ is implemented using a multimodal large language model (MLLM) that analyzes generated images and outputs semantically precise textual descriptions, which serve as proxies for the model's preferred prompts.

## 4.3 INFORMATION ENHANCER

The **information enhancer** learns to map user prompts $P_{\text{user}}$ into the preference-aligned space by using LLMs; details are shown in Section 5.1 and Figure 3. Formally, given model parameters $\theta$, the enhancer generates a distribution $p_\theta(\cdot \mid P_{\text{user}})$ that approximates $\mathcal{D}_{\text{pref}}$. The training loss is defined as

$$\mathcal{L}(\theta) = \sum_{t=1}^{T} \mathbb{E}_{x_{<t} \sim p_\theta} \Big[ KL\big( p_{\theta,t}(\cdot \mid x_{<t}, P_{\text{user}}) \,\|\, \mathcal{D}_{\text{pref}} \big) \Big],$$

which can be equivalently approximated as a cross-entropy objective:

$$\hat{\mathcal{L}}(\theta) = -\mathbb{E}_{(P_{\text{user}}, I)} \Big[ \mathbb{E}_{x \sim \mathcal{D}_{\text{pref}}} \Big[ \sum_{t=1}^{T} \log p_{\theta,t}(x_t \mid x_{<t}, P_{\text{user}}) \Big] \Big].$$

## 4.4 INFORMATION ALIGNER

While the enhancer promotes alignment with $\mathcal{D}_{\text{pref}}$, it does not explicitly enforce semantic consistency between visual entities in the generated image and the prompt. To address this, we introduce the **information aligner**, which augments the enhancer with entity-level alignment. Specifically, we extract salient entities from the user prompt and generated; more details are shown in Section 5.1:

$$P_{\text{entity}} = g_{\text{entity}}(P_{\text{user}}), \quad P_{\text{entity}} = g_\phi(\mathcal{G}(P_{\text{user}})),$$

where $\mathcal{G}$ denotes the generative model. The enhanced prompt is then formed as

$$\tilde{P} = \text{Concat}(P_{\text{user}}, P_{\text{entity}}).$$

The training loss with the aligner is defined as:

$$\hat{\mathcal{L}}(\theta) = -\mathbb{E}_{(P_{\text{user}}, I)} \left[ \mathbb{E}_{x \sim \mathcal{D}_{\text{pref}}} \left[ \sum_{t=1}^{T} \log p_{\theta, t}(x_t \mid x_{<t}, \tilde{P}) \right] \right].$$

During inference, directly invoking $\mathcal{G}$ for entity extraction is computationally expensive. As shown in Figure 3, to balance efficiency and performance, we adopt a lightweight surrogate generator $\mathcal{G}'$, which produces a fast pre-generation $\hat{I} = \mathcal{G}'(P_{\text{user}})$. Entities are extracted from $\hat{I}$ and concatenated with the original user prompt before enhancement. This strategy ensures both high generation fidelity and practical inference efficiency.

## 5 EXPERIMENTS

### 5.1 SETTINGS

We conduct experiments using the publicly available text-to-image model *Stable Diffusion 2.1* (SD 2.1) (Rombach et al., 2022) and its enhanced variant *CoMat* (Jiang et al., 2024), evaluating the impact of different prompt optimization methods on both image quality and generation stability. For the LLM backbone, we employ *Mistral-7B-Instruct-v0.2* (Jiang et al., 2023), fine-tuned to act as an *information enhancer*. The *information aligner* is implemented via *InternVL3-2B* (Zhu et al., 2025), which extracts entities during training. During inference, users can either explicitly specify entities or rely on *InstaFlow* (Liu et al., 2023) to generate candidate entities, which are subsequently extracted using *InternVL3-2B*.

Training details are shown in Appendix C.1. To model user preference distributions, we leverage *Gemini 2.0* (Team et al., 2025) for high-fidelity textual descriptions. The *DiffusionDB* dataset (Wang et al., 2022) provides a diverse set of text-image pairs, from which 1,000 pairs are randomly sampled for fine-tuning. Evaluation is performed on multiple benchmark datasets, including *Auth-Prompt Bench* and *T2I-CompBench* (Huang et al., 2023), covering a wide range of prompt styles and complexities. Metrics include prompt and image stability, relevance, and diversity. All experiments are conducted on NVIDIA A100 40GB GPUs and the same seed 995 to ensure reproducibility and fair comparison. The overview of **NoxEye** are in Figure 3.

### 5.2 COMPARATIVE METHODS

We compare our approach against *Promptist* (Hao et al., 2023), and the text-to-image model use *SD 2.1* (Rombach et al., 2022) and *CoMat 2.1* (Jiang et al., 2024). *Promptist* relies on a pre-trained language model, while *CoMat* incorporates multimodal information for prompt refinement. Performance is evaluated across different base text-to-image models (*SD 2.1* and *CoMat*) and various method-model combinations. For fairness, *Promptist* uses its publicly released model, whereas *CoMat*, without a public version, is trained under the same experimental settings.

### 5.3 RESULTS

**Auth-Prompt Bench** As shown in Table 2, our method consistently surpasses existing generative models and prompt-optimization baselines across all evaluation metrics and prompt categories.

Table 2: Evaluation results on Auth-Prompt Bench.

| METHOD | ACCURACY ↑ | | ENTROPY ↓ | | ENERGY ↓ | |
|---|---|---|---|---|---|---|
| | novice | expert | novice | expert | novice | expert |
| SD 2.1 | 20.80% | 30.16% | 2.8129 | 2.5309 | -10.5735 | -12.0765 |
| CoMat 2.1 | 20.59% | 29.83% | | | | |
| Promptist+SD 2.1 | 1.12% | 1.12% | 2.3339 | 2.3038 | -12.6654 | -12.3914 |
| Promptist+CoMat 2.1 | 1.70% | 1.54% | | | | |
| Ours+SD 2.1 | 19.31% | 30.94% | 2.3136 | 2.1674 | -13.1233 | -13.8620 |
| Ours+CoMat 2.1 | 19.19% | 31.06% | | | | |
| Ours*+SD 2.1 | **34.32%** | **34.88%** | 2.2420 | 2.0802 | -13.4299 | -14.2882 |
| Ours*+CoMat 2.1 | 33.44% | 34.30% | | | | |

Note: Our* refers to the user-specified entity. The best result is bolded.

Table 4: Impact of finetune information enhancer and information aligner.

| finetune | aligner | ACCURACY ↑ | | ENTROPY ↓ | | ENERGY ↓ | |
|---|---|---|---|---|---|---|---|
| | | novice | expert | novice | expert | novice | expert |
| | | 19.62% | 30.28% | **2.1834** | 2.0850 | **-13.7830** | -14.2698 |
| ✓ | | 19.33% | 29.71% | 2.1876 | 2.0857 | -13.7671 | -14.2720 |
| ✓ | unspecified | 19.31% | 30.94% | 2.3136 | 2.1674 | -13.1233 | -13.8620 |
| | specified | 33.58% | 34.82% | 2.2415 | 2.0835 | -13.4243 | -14.2837 |
| ✓ | specified | **34.32%** | **34.88%** | 2.2420 | **2.0802** | -13.4299 | **-14.2882** |

Note: The best result is bolded. 'Specified' or 'Unspecified' indicates whether the user has specified the entity.

Table 3: Evaluation on T2I-CompBench.

| Model | 2D-Spatial ↑ | 3D-Spatial ↑ | Non-Spatial ↑ | Numeracy ↑ |
|---|---|---|---|---|
| SD 2.1 | 0.0714 | 0.2239 | 0.2979 | 0.3811 |
| CoMat 2.1 | 0.0754 | 0.2204 | **0.2987** | 0.3725 |
| Promptist+SD 2.1 | 0.0528 | 0.2207 | 0.2831 | 0.3632 |
| Promptist+CoMat 2.1 | 0.0519 | 0.1913 | 0.2790 | 0.3520 |
| Ours+SD 2.1 | 0.0770 | 0.2324 | 0.2938 | 0.3852 |
| Ours+CoMat 2.1 | **0.0793** | **0.2346** | 0.2949 | **0.3976** |

Note: The best result is bolded. For additional results, see Appendix C.2.

Table 5: Impact of preference distribution extractor.

| EXTRACTOR | ACCURACY ↑ | | ENTROPY ↓ | | ENERGY ↓ | |
|---|---|---|---|---|---|---|
| | novice | expert | novice | expert | novice | expert |
| Gemini 2.0 Flash | 34.32% | 34.88% | 2.2420 | 2.0802 | -13.4299 | -14.2882 |
| GPT-4o-mini | 34.32% | 34.88% | **2.2401** | **2.0801** | **-13.4318** | **-14.2893** |

Note: The best result is bolded.

Table 6: Impact of Enhancer Backbone.

| Enhancer Backbone | ACCURACY ↑ | | ENTROPY ↓ | | ENERGY ↓ | |
|---|---|---|---|---|---|---|
| | novice | expert | novice | expert | novice | expert |
| Mistral-7B-Instruct-v0.2 | **34.32%** | **34.88%** | **2.2420** | **2.0802** | **-13.4299** | **-14.2882** |
| Llama-3.1-8B-Instruct | 10.00% | 8.58% | 3.1487 | 3.1680 | -9.8730 | -9.7044 |

Note: The best result is bolded.

On the novice split, *Ours*+SD 2.1* improves accuracy from 20.80% (SD 2.1) to 34.32%, while *Promptist+SD 2.1* achieves only 1.12%. The reductions in prompt entropy (from 2.8129 to 2.2420) and prompt energy (from -10.5735 to -13.4299) indicate improved stability and higher fidelity of the generated images. On the expert split, accuracy increases from 30.16% to 34.88%, with entropy and energy decreasing from 2.5309 and -12.0765 to 2.0802 and -14.2882, respectively. In contrast, improvements by Promptist over the baseline are marginal, highlighting the robustness of our method across both simple and complex prompts.

We found that Promptist's low accuracy primarily results from text degeneration and repetition. First, it uses GPT-2 (Radford et al., 2019) as its backbone, which has a maximum context length of 1024 tokens—shorter than many authentic prompts, limiting its ability to fully capture their semantics. Second, GPT-2's relatively small parameter size makes it inherently prone to repetitive content (Li et al., 2023). Third, its training setup exacerbates the problem: the main content of real user prompts was used as input, and the original prompts as output, biasing the model toward adding superficial modifiers rather than performing meaningful optimization (Yao et al., 2025).

**T2I-CompBench** We further evaluate generalization across different text-to-image backbones. As reported in Table 3, *Ours+CoMat 2.1* outperforms the baseline in multiple capability dimensions, including 2D spatial reasoning (0.0793), 3D spatial reasoning (0.2346), and numeracy (0.3976). In comparison, Promptist often exhibits negligible or inconsistent gains. Although our method performs slightly below the baseline on attribute binding and complex compositions, it remains competitive with or superior to Promptist. These results suggest that our approach enhances compositional understanding, structural reasoning, and numerical alignment in generated images.

## 5.4 ABLATION STUDY

To assess the contributions of the *information enhancer* and *information aligner* to generation quality and stability, we conduct a series of ablation experiments. First, we isolate the effect of the information enhancer. As shown in Table 4, compared to the baseline without enhancement, accuracy remains essentially unchanged, while both prompt entropy and energy decrease. Next, we evaluate the information aligner. Without specifying entities, the aligner increases accuracy on the expert dataset but decreases it on the novice dataset, reflecting the higher ambiguity of novice prompts.

We then test the impact of the *preference extractor* by replacing Gemini 2.0 with GPT-4o-mini OpenAI et al. (2024). Results show negligible differences in accuracy, energy, and entropy, while Gemini 2.0 offers a faster response time; hence, we adopt Gemini 2.0 as the extractor.

Finally, we investigate the choice of backbone for the enhancer (Table 6). Replacing Mistral-7B-Instruct-v0.2 with Llama-3.1-8B-Instruct (Meta AI, 2024) consistently degrades performance across all metrics. We attribute this to catastrophic forgetting observed during training (Kirkpatrick et al., 2017).

Table 7: Inference time comparison across different methods.

| Method | SD 2.1 | Promptist+SD 2.1 | Mistral-7B+SD 2.1 | Ours+SD 2.1 | Ours*+SD 2.1 |
|---|---|---|---|---|---|
| **Inf. Time (s)** | 0.0327 | 0.0358 | 0.0748 | 1.9549 | 0.0782 |

## 5.5 HUMAN EVALUATION

We conducted a user study with 20 volunteers to compare our method with existing approaches from a human-centered perspective. Our approach was preferred most often, achieving scores of 0.44 (images) and 0.412 (prompts), compared to 0.417/0.143 and 0.366/0.222 for the baselines, respectively, demonstrating our method's superior alignment with human preferences (see Appendix C.3).

## 5.6 INFERENCE TIME COMPARISON

We further compare the inference efficiency of different prompt optimization methods. Results show that when user-specified entities are provided, our method achieves comparable inference time to Promptist and CoMat, with differences within 0.1s. By contrast, approaches relying on InstaFlow for pre-generation combined with InternVL3 for entity extraction incur longer latency, though still under 2s. Notably, while our method exhibits a slight increase in inference time for highly complex prompts, the overall cost remains well within a practical range. Quantitative results are summarized in Table 7.

## 5.7 QUALITATIVE RESULTS

Figure 4 illustrates additional visual examples. Our method effectively refines the main content and provides detailed descriptions of artistic style, lighting, and other visual attributes.

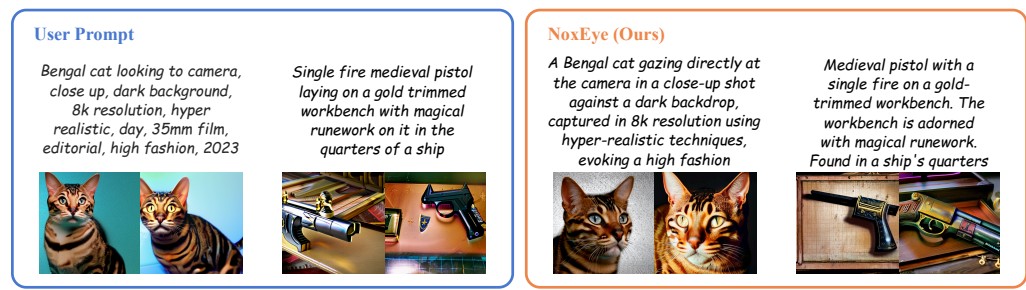

Figure 4: The generated images with the optimized prompts using our method. Each prompt generates two images, the left one uses SD 2.1 (Rombach et al., 2022) and the right one uses CoMat 2.1 (Jiang et al., 2024). More results are in the Appendix D.

## 6 CONCLUSION

In this work, we revisit the problem of information transfer in text-to-image generation and propose new measures to directly quantify the stability between user intent and generated outputs. To this end, we construct **Auth-Prompt Bench**, a benchmark consisting of novice and expert prompts, and introduce three complementary metrics—classification accuracy, prompt entropy, and prompt energy—to evaluate stability. Building on these insights, we design **NoxEye**, a plug-and-play modular prompt optimization framework broadly compatible with diverse T2I diffusion models. NoxEye combines an *information enhancer*, which leverages large language models to enrich prompt semantics, and an *information aligner*, which aligns visual concepts with user intent via multimodal grounding. Extensive experiments across Auth-Prompt Bench and additional benchmarks demonstrate that our approach substantially improves both image quality and stability, without introducing significant inference overhead. Overall, our contributions provide not only a practical method for prompt optimization, but also a novel perspective on modeling information flow in text-to-image generation. We believe this work lays a foundation for future research on principled evaluation and optimization of user–model interactions in generative systems.

ETHICS STATEMENT

All prompts and images were curated to ensure ethical compliance and user safety, with manual checks to maintain expertise levels and exclude harmful and insafe content. Data sources, `Lexica` (`https://lexica.art/`) and `Civitai` (`https://civitai.com/`), are publicly accessible, and no personally identifiable information was used. Training used the `DiffusionDB` (Wang et al., 2022) dataset (CC0 1.0 License), fully complying with licensing terms. This protocol upholds ethical standards for both participants and model-generated content consumers.

REPRODUCIBILITY STATEMENT

To ensure reproducibility, we provide the dataset, source code, and intermediate outputs in the supplementary materials. After paper acceptance, we will open-source them on GitHub. More experiment settings are in Apendix C.1. In summary, we have made every effort to ensure the reproducibility of this paper.

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

## A LLM Usage

Large Language Models (LLMs), specifically GPT and Gemini, were employed solely for language polishing and minor stylistic refinement of this manuscript. The models were not involved in research ideation, experiment design, data analysis, or substantive content generation. Their role was limited to improving clarity, grammar, and formatting of text written by the authors.

## B Additional Details on Stability Modeling

### B.1 Intent Distribution Approximation

For a given entity $e \in \mathcal{E}$ and a set of templates $\mathcal{T} = \{t_k\}_{k=1}^{K}$, we define the approximate intent distribution as

$$p(y \mid e) = \frac{1}{K} \sum_{k=1}^{K} \delta(y - t_k(e)),$$

where $\delta(\cdot)$ is the Dirac delta. This approximation underlies the mutual information lower bound in the main text:

$$I(Y;I) \approx H(Y) + \mathbb{E}_{I,e,y} \log q(y \mid I), \quad y \sim p(y \mid e),$$

linking image–text consistency to the likelihood of recovering a distribution of template-based prompts representing the entity concept.

### B.2 Fano's Inequality

Fano's inequality bounds conditional entropy in terms of classification error $e = \Pr(\hat{Y} \neq Y)$:

$$H(Y \mid I) \leq H(e) + e \log(K - 1),$$

where $H(e) = -e \log e - (1 - e) \log(1 - e)$. Assuming uniform prior $H(Y) = \log K$, this yields a lower bound on mutual information:

$$I(Y;I) \geq \log K - H(e) - e \log(K - 1),$$

showing that higher intent-recovery accuracy implies greater prompt stability.

### B.3 Prompt Entropy and Information-Theoretic Derivation

Consider the Markov chain $Y \to P \to I$, where $Y$ is user intent, $P$ is the prompt, and $I$ is the generated image. By the data-processing inequality:

$$I(Y;I) \leq I(Y;P),$$

indicating that the maximum achievable stability is constrained by the prompt information content.

Operationally, for a prompt $P = (w_1, \ldots, w_T)$, we approximate entropy using a pretrained LM:

$$H(P) \approx -\frac{1}{T} \sum_{t=1}^{T} \log p_\theta(w_t \mid w_1, \ldots, w_{t-1}).$$

**Interpretation:**

- *Low cross-entropy:* Predictable, concentrated prompt effectively conveys user intent, enhancing stability.
- *High cross-entropy:* Uncertain or dispersed prompt, less informative, reducing stability.

### B.4 Derivation of Prompt Energy

In classical statistical mechanics, a system state $x_t^{(i)}$ follows a Boltzmann distribution:

$$p(x_t^{(i)}) = \frac{\exp(-E_t^{(i)}/k\tau)}{Z_t}.$$

An autoregressive LM with parameters $\theta$ defines the probability of token $x_t$ as

$$p_\theta(x_t \mid x_{<t}) = \frac{\exp(z_t(x_t))}{\sum_{v \in \mathcal{V}} \exp(z_t(v))},$$

where $z_t(v)$ is the logit of token $v$.

Identifying logits with negative energies up to a normalization constant $C_t$:

$$z_t(v) = -\frac{1}{k\tau} E_t(v) + C_t.$$

Setting $k\tau = 1$ and $C_t = 0$ yields token-level energy

$$e_t := E_t(x_t) = -z_t(x_t),$$

and sequence-level prompt energy

$$E(x) = -\frac{1}{T} \sum_{t=1}^{T} z_t(x_t),$$

which measures the model's confidence in generating $x$. Lower $E(x)$ indicates familiar, well-represented concepts, whereas higher $E(x)$ indicates uncertain or out-of-distribution concepts.

**Usage.** Prompt energy complements entropy and end-to-end mutual information metrics, enabling a more complete characterization of text-to-image generation stability.

## C MORE EXPERIMENT RESULTS

### C.1 EXPERIMENTAL SETUP IMPLEMENTATION DETAILS

**Training Hyperparameters Settings.** We trained our model with the following hyperparameters: a learning rate of $1 \times 10^{-5}$, a batch size of 2, gradient accumulation steps of 16, and a total of 3 training epochs. The checkpoint with the lowest training loss was selected as the final model.

During LoRA fine-tuning, all parameters of the base model were frozen, and only the LoRA parameters were updated, specifically for query and value (Vaswani et al., 2017; Hu et al., 2022). The LoRA hyperparameters were set as follows: rank $r = 8$, $\alpha = 16$, a dropout rate of 0.1, and no bias. Training was performed using bf16 mixed precision.

We employed the Adam optimizer with $\beta_1 = 0.9$, $\beta_2 = 0.999$, and a weight decay of $1 \times 10^{-2}$.

**NoxEye Prompt Template.** To ensure consistency in model evaluation, we adopt the NoxEye Prompt Template, which specifies a unified structure for presenting tasks, inputs, entity and outputs. The template is organized into four components:

- *Instruction*: defines the task description or objective to be performed.
- *Input*: provides the contextual information or query required to complete the task.
- *Main Object*: highlights the key entities that are central to the task.
- *Response*: represents the expected model-generated output.

When the user does not specify any entities, we first use InstaFlow to quickly generate the image, and then employ InternVL3-2B to extract entities. The prompt template used by InternVL3-2B to extract the main entities from the image is illustrated in the Figure 5.

**Evaluation Settings.** For evaluation on **Auth-Prompt Bench**, the following hyperparameters were used: the number of sampling steps was set to 50, the CFG scale to 7.5, the image size to $512 \times 512$, and the batch size to 20. For evaluation on **T2I-CompBench**, the hyperparameters were set as follows: the number of sampling steps was 50, the CFG scale 7.5, the image size $512 \times 512$, and the batch size 1, with 4 images generated per prompt.

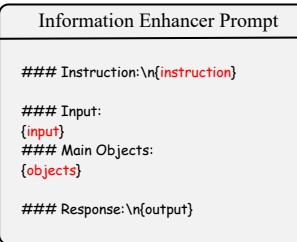 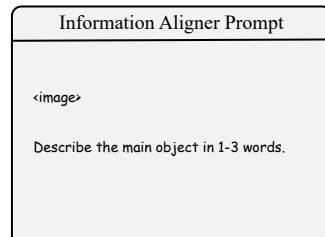

Figure 5: The prompt template of information enhancer and information aligner.

**Preference Distribution Extractor Setting.** We present the prompt to transform the LVLM into the distribution extractor. The prompt for the preference distribution extractor in our model is illustrated in the Figure 6. As shown in Figure 7, the distribution of authentic prompts is significantly higher than that of prompts extracted by the preference distribution extractor, indicating that the latter produces prompts that are more concise and stable.

---

**Preference Distribution Extractor Prompt**

You are a professional AI image analyst specializing in analyzing Stable Diffusion generated images. Please analyze this image and generate a prompt that could have been used to create this image.

Requirements:
1. The generated prompt should be concise, accurate, and suitable for CLIP model understanding
2. Use English with comma-separated keyword format
3. Include the following elements (if applicable):
   - Subject description (people, objects, scenes)
   - Art style (e.g., realistic, anime, oil painting, digital art, etc.)
   - Quality descriptors (e.g., highly detailed, 8k, masterpiece, etc.)
   - Composition description (e.g., portrait, full body, close-up, etc.)
   - Lighting effects (e.g., soft lighting, dramatic lighting, etc.)
   - Color characteristics (e.g., vibrant colors, monochrome, etc.)

4. Avoid overly complex descriptions, keep the prompt practical
5. Sort by importance, with the most important keywords first

Please output the prompt directly without additional explanations.

---

Figure 6: Preference distribution extractor prompt template.

## C.2 FULL RESULTS OF T2I-COMPBENCH

In Table 8, our method (Ours+CoMat 2.1) demonstrates clear advantages in several challenging dimensions. While CoMat 2.1 (Jiang et al., 2024) achieves the highest scores on basic visual attributes such as color, shape, and texture, integrating our optimization yields the best overall performance on spatial reasoning tasks, achieving 0.0793 for 2D-Spatial and 0.2346 for 3D-Spatial, corresponding to relative improvements of approximately 5.2% and 6.4% over the CoMat 2.1 baseline.

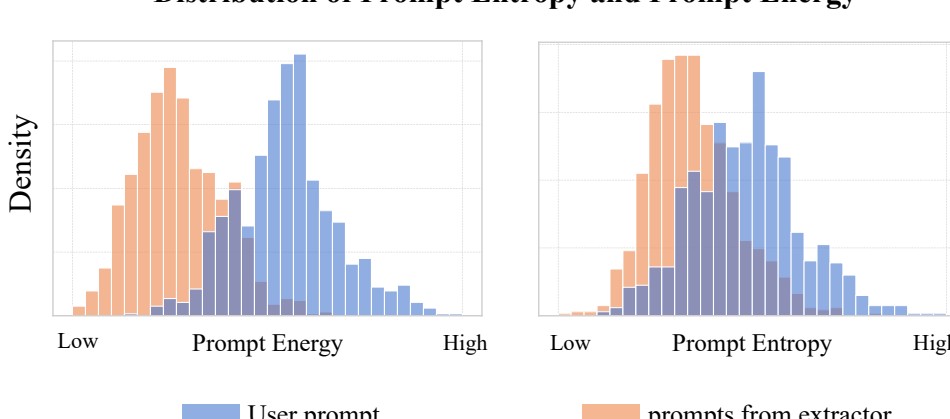

Figure 7: A visual comparison between authentic prompts and prompts from extractor.

Table 8: Evaluation on T2I-CompBench.

| Model | Color | Shape | Texture | 2D-Sp | 3D-Sp | Non-Sp | Numeracy | Complex |
|---|---|---|---|---|---|---|---|---|
| SD 2.1 | 0.3988 | 0.2981 | 0.2768 | 0.0714 | 0.2239 | 0.2979 | 0.3811 | 0.2966 |
| CoMat 2.1 | **0.4274** | **0.3292** | **0.3266** | 0.0754 | 0.2204 | **0.2987** | 0.3725 | **0.3043** |
| Promptist+SD 2.1 | 0.3653 | 0.2840 | 0.2388 | 0.0528 | 0.2207 | 0.2831 | 0.3632 | 0.2725 |
| Promptist+CoMat 2.1 | 0.3931 | 0.3190 | 0.2961 | 0.0519 | 0.1913 | 0.2790 | 0.3520 | 0.2721 |
| Ours+SD 2.1 | 0.3462 | 0.2675 | 0.2448 | 0.0770 | 0.2324 | 0.2938 | 0.3852 | 0.2856 |
| Ours+CoMat 2.1 | 0.3823 | 0.2983 | 0.3189 | **0.0793** | **0.2346** | 0.2949 | **0.3976** | 0.2959 |

Note: The best result is bolded.

Furthermore, Ours+CoMat 2.1 achieves the highest numeracy score (0.3976) among all compared methods, indicating superior handling of quantitative concepts. Importantly, these gains are obtained without substantial degradation in low-level visual fidelity, contrasting with Promptist (Hao et al., 2023), which exhibits performance drops in color and texture when attempting to improve spatial understanding.

Overall, these results highlight that our approach significantly enhances high-level semantic alignment and spatial reasoning while maintaining balanced performance on basic perceptual attributes. This suggests that our method provides a more robust strategy for text-to-image prompt optimization in complex compositional scenarios.

### C.3 HUMAN EVALUATION RESULTS

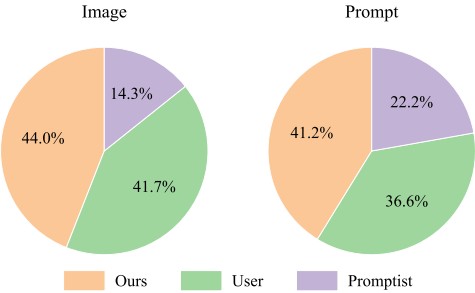

Figure 8: Human evaluation results. The result of NoxEye are preferred by human compared with the result of User Prompt and Promptist (Hao et al., 2023).

To complement quantitative evaluations with a human-centered perspective, we conducted a user study comparing our method with existing approaches. We first sampled a set of prompts at random

and applied different optimization methods to obtain model-specific refined prompts, which were then used to generate images. A total of 20 volunteers were recruited from diverse educational backgrounds. In each trial, participants were presented with either a pair of images or a pair of prompts and were asked to select the image they found more visually appealing or the prompt they preferred. As shown in Figure 8, participants most frequently selected images generated from prompts optimized by our method, indicating its superior effectiveness in aligning with human preference.

# D MORE QUANTITATIVE RESULTS

We present more visual results between the images generated with different prompts. As shown in 9, the optimized prompts result in more pleasing images. A more intuitive observation is that the flat, uninteresting view in Minecraft and the more aesthetically pleasing, more detailed view are represented by the optimized prompt before and after optimization, respectively. In addition, the modified prompt has stronger alignment capabilities. For example, the prompt "Honeybee by peter paul rubens" is amended to "A honeybee painting by Peter Paul Rubens".

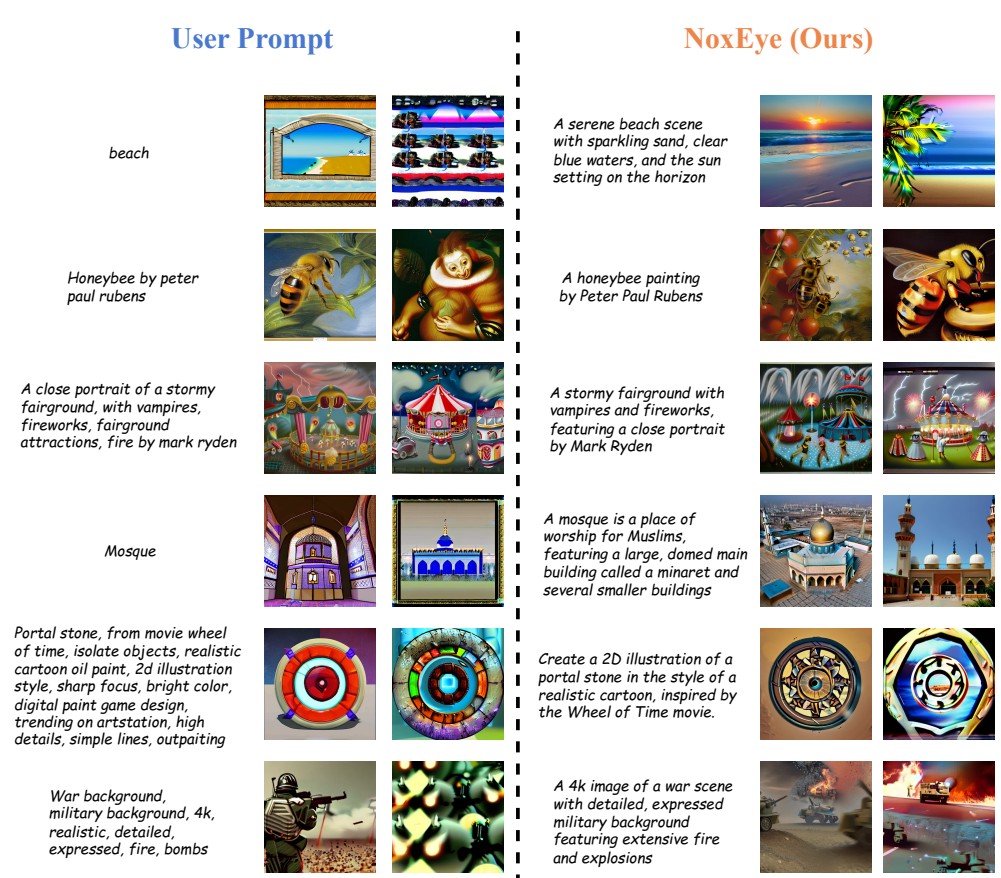

Figure 9: The images generated using Noxeye. Each prompt generates two images, the left one uses SD 2.1 (Rombach et al., 2022) and the right one uses CoMat 2.1 (Jiang et al., 2024).

