# OpenReview forum: "Auth-Prompt Bench: Towards Reliable and Stable Prompting in Text-to-Image Generation"
_ICLR.cc/2026/Conference — ICLR 2026 Conference Withdrawn Submission_

### Official Review · Reviewer_qybK · 2025-10-16

**Soundness:** 2
**Presentation:** 3
**Contribution:** 2
**Rating:** 2
**Confidence:** 5

**Summary:**

This work introduces information-theoretic measures to study text-to-image generative models, with the goal of improving generation "stability" through an elaborate method that transforms user prompts into preferable and stable prompts that produce "better" images.

The proposed method consists in several components, all relying on "external" language and multimodal models to extract an ineal preference distribution of prompt elements from the perspective of the T2I model, and an alignment module that learns the parameters of a model to transform user prompts into new prompts that are more aligned to the preference distribution.
Intuitively, the overall scheme attempts at improving the mutual information between user intent and the obtained image.

The authors evaluate their method on an original benchmark proposed by the authors, as well as on a standard benchmark to quantify T2I alignment. The proposed method is compared to Promptist, CoMat and vanilla Stable Diffusion. Results indicate that the proposed method outperforms alternatives on several metrics. An ablation study is also performed to understand the contribution of each proposed component toward the superior performance measured in the experiments.

**Strengths:**

* The study of alignment is central in modern generative models, and this work addresses an important problem in the context of text-to-image generation.

* The proposed information-theoretic measures are a nice alternative to heuristics that are often found in related work.

* The authors propose a new benchmark, with a collection of user intent, which is quite unique and useful.

**Weaknesses:**

* The proposed method should be clarified, as it might suffer from circular reasoning. In short (but see detailed questions below), the authors should provide an intuition about wether transforming user prompts into those that match a preferred distribution according to a variety of external model internal representations does not affect negatively the very metric they wish to optimze, the mutual information between intent and generated image.
The fact that performance tables (and user studies) indicate superiority of the proposed method should be taken with a grain of salt: metrics used in existing benchmarks such as T2I-CompBench are based on the very same models the authors use to optimze alignment to intent, so the obtained results might be biased.

* The authors have neglected related work that also propose information theoretic measures to improve T2I models [1]. Although authors in [1] discuss a fine-tuning method, the principle of aligning user prompts to images through the lenses of mutual information has been done before, and should be discussed.

* The experimental validation lacks details about performance metrics. This is especially true for the T2I-CompBench evaluation: the authors of that work use several auxiliary models to measure performance, some of which might coincide with the models used in this work to optimize intent to image generation. As such results might be biased.

* Recent developments in T2I models include flow-based generative models, as well as autoregressive methods, which appear to enjoy improved quality and alignment. I believe the authors should discuss these models, and indicate whether their proposed method can be applied in those practical cases.

[1] Wang, C, et al. "Information Theoretic Text-to-Image Alignment", ICLR 2025

**Questions:**

* In line 142-143 you define prompt stability through the lenses of the mutual information between the distribution of a random variable representing user intents, and the distribution of a random variable representing the associated generated image. In line 245-246 you indicate that the combination of the three metrics defined in this work, MI, prompt entropy and energy is used to quantify generation stability. This is confusing: is there a difference between prompt and generation stability? Where did you define generation stability? If the two are the same, then it looks like a circular argument.

* In line 254-255 you indicate that the proposed metrics are aggregated at instance and class levels. Can you clarify what is an instance (a single image, I imagine) and a class (intent category, I guess?).

* In section 4.1, line 294 you define the transformation (a model that you have to learn), to map the distribution of user prompts to a distribution of stable prompts. Then in line 295 you define $p_{\theta} (P*)$, but do not define $P*$. Also, you do not define what exactly is the ideal preference distribution $\mathcal{D}_{\text{pref}}$.

* Again in section 4.1, line 298-299, the optimal parameters $\theta^*$ are obtained through a distance measure, but the arguments of the distance do not match with the description. In the expression, you have a distance between the prompt $P_{\text{stable}}$, and a preference distribution $\mathcal{D}$, which is averaged over all user prompts, and the associated image. Where does the associated image appear in the expression? Is $\mathcal{D}_{\text{pref}}$ dependent on the image associated to the transformed user prompt? Could you please clarify?

* In section 4.2 you clarify how to obtain $\mathcal{D}_{\text{pref}}$. Would you mind moving this definition before the method to learn a model to map user prompts to a stable prompt? Also, more importantly, what you are doing here, to my understanding, is as follows. Use an external multimodal LLM to map an image to a preferred distribution, which is a textual description of the image. Then you learn a model that use such learned distribution, to learn a model that maps a user prompt to make it closer to the learned, preferred distribution. Finally, you use another model to learn judgment metrics (prompt entropy and energy with Gemma, MI with CLIP). Isn't there a risk to push user prompts to comply to internal representations of external models (Gemma and CLIP), thus defeating the whole purpose of making sure user intent is understood by the T2I model and transformed in an appropriate generated image?

* In Table 2, can you please define accuracy?

* In Table 3, can you define the metrics you chose from T2ICompBench? Is this a CLIP score? BLIP-VQA? Human preference score?

---

### Official Review · Reviewer_1S62 · 2025-10-29

**Soundness:** 2
**Presentation:** 1
**Contribution:** 2
**Rating:** 4
**Confidence:** 3

**Summary:**

This paper introduces a benchmark of 17580 prompts, named the Authentic Prompt Benchmark (Auth-Prompt Bench), to quantify the stability of the prompts in text-to-image generation. The work focuses on using metrics from information theory to evaluate the prompt-to-prompt transmission, including mutual information, prompt entropy, and prompt energy. Further, this work also contributes an end-to-end prompt optimization framework including an information enhancer to map user prompts toward the model-preferred distribution with LLMs and an information aligner to enforce fine-grained alignment of visual entities between the images generated by the original and enhanced prompts. The proposed framework achieved enhanced performance in image quality and stability on the proposed benchmark and other existing benchmarks.

**Strengths:**

1. This paper contributes a framework to enhance the prompt and a benchmark with three metrics to evaluate the prompt stability in text-to-image generation. The proposed framework achieved great performance and has potential in practical applications.

2. The design of the framework focuses on both enhancing the text prompt and aligning the visual entity, which is well-considered in the text-to-image scenario.

3. The experiments evaluate multiple aspects, including various text-to-image models and LLMs. Evaluation is also comprehensive, with the results on another benchmark, inference time, and user study.

**Weaknesses:**

1. Instability or expected diversity. The paper claimed that "ambiguous or under-specified prompts often lead to unstable outputs that deviate from user intent, while current benchmarks provide limited means to quantify this phenomenon" and defines this setup as the instability of prompting. Though I agree that augmenting the prompts can be practically useful, under-specified prompts might reflect a user intent of expecting more diversity in the generated images.

2. Lack of clarity. The notions in Section 3.1 are used without definition or proper discussion, for example:

   2.1 The notions of $Y, P, I$ in line 141, $Y(e)=\\{ t_k(e) \\}_{k=1}^K$ in line 152, $H(e)$ in line 159, $w_t$ in line 180, are used without any explanation, making the presentation of this section confusing.

   2.2 Some of the notations such as $p(y|I), q(y|I)$ or $\\phi_{img}(I), \\phi_{text}(y)$, though consistent and understandable in the context of text-to-image research, adding brief definitions could make this work more accessible to a broader audience.

   2.3 The aligner seems to apply to the generated images in Figure 3, while the discussion in Section 4.4 describes it as extracting from user and generated prompts.

3. The benchmark evaluation primarily focuses on the Promptist work, which might be limited for further application. A few works of augmenting the prompts for text-to-image models might need to be included too, such as [1, 2].

[1] Mo, Wenyi, et al. "Dynamic prompt optimizing for text-to-image generation." Proceedings of the IEEE/CVF Conference on Computer Vision and Pattern Recognition. 2024.
[2] Datta, Siddhartha, et al. "Prompt expansion for adaptive text-to-image generation." arXiv preprint arXiv:2312.16720 (2023).

**Questions:**

1. The work focuses on augmenting and evaluating the textual prompt. Are the proposed metrics applicable to the methods of tuning the prompt embeddings?

2. Following the weakness 2.3 above, assuming the aligner uses the images to extract entities. As the paper claims in the introduction, the images generated from the original prompt might be unstable. In that case, why does the aligner aim to align the visual entities from the generated images rather than the user prompt?

3. Minor: How to make sure that the augmented prompts don't introduce details that are not specified in the original prompt?

---

### Official Review · Reviewer_F2SN · 2025-10-31

**Soundness:** 2
**Presentation:** 2
**Contribution:** 2
**Rating:** 2
**Confidence:** 4

**Summary:**

- This paper constructs Auth-Prompt Bench, a benchmark comprising 17,580 real-world prompt-image pairs for evaluating generation stability in text-to-image (T2I) generation, with prompts stratified into novice and expert categories. The authors also propose NoxEye, an information-theoretic end-to-end prompt optimization framework that achieves improvements of up to 13.52% in mutual information, 20.30% in prompt entropy, and 27.01% in prompt energy.

**Strengths:**

- **Introduction of Information-Theoretic Evaluation Metrics**: While existing metrics focus solely on prompt-image alignment, this work establishes a foundation for quantifying the stability of prompt-to-prompt transmission from an information-theoretic perspective by introducing three complementary metrics: Mutual Information, Prompt Entropy, and Prompt Energy.

- **Large-Scale Benchmark Based on Authentic Prompts**: The benchmark collects 17,580 authentic prompts from Lexica (novice users) and Civitai (expert users), providing a systematic environment for validating the relationship between user proficiency, prompt informativeness, and generation instability.

- **Generalizability and Practical Efficiency**: NoxEye adopts a plug-and-play modular design applicable to diverse T2I diffusion models such as Stable Diffusion 2.1 and CoMat. The reported inference time when using user-specified entities is comparable to Promptist (within 0.1 seconds difference).

**Weaknesses:**

- **W1: Gap Between Theoretical Framework and Implementation**
    - While this work is motivated by maximizing mutual information $I(Y;I)$, the "preference distribution" $\mathcal{D}_{\text{pref}}$ used in actual training is approximated by high-fidelity textual descriptions generated by an MLLM (Gemini 2.0). There is no guarantee that this distribution truly represents user intent or the optimal distribution for the diffusion model. Consequently, it remains unclear whether the optimization actually corresponds to maximizing $I(Y;I)$.

- **W2: Model Dependency of Evaluation Metrics**
    - Mutual information is evaluated via CLIP-based zero-shot classification, while prompt entropy and energy are measured using Gemma-3-1B. These metrics strongly depend on the behavior of the employed models (CLIP/Gemma), and evaluation results may vary when different language models or image-text models are used.

- **W3: Scale and Representativeness of Human Evaluation**
    - The human evaluation was conducted with 20 volunteers. To make strong claims about subjective superiority across diverse prompts and domains, the sample size and participant diversity appear insufficient.

**Questions:**

**Q1**: Regarding the preference distribution $\mathcal{D}_{\text{pref}}$ being generated by an MLLM (Gemini 2.0), how did you validate that this distribution truly represents user intent or the optimal distribution for the diffusion model?

**Q2**: Given that the evaluation of mutual information, entropy, and energy depends on CLIP and Gemma-3-1B respectively, how do you consider the stability of evaluation when other models (e.g., alternative LLMs or image encoders) are employed?

---

### Note · Authors · 2026-04-01

I have read and agree with the venue's withdrawal policy on behalf of myself and my co-authors.

---

### Meta-Review · Area_Chair_ofpC · 2026-01-03

**Summary:**

The manuscript is reviewed by three reviewers with all the reviewers generally on the negative side highlighting several issues in the paper including, a gap between the theoretical framework and actual method implementation, insufficient scale and representativeness of human evaluation, lack of clarity in the manuscript (e.g., notions in Section 3.1 are used without definition or proper discussion), limited future potential of the benchmark, missing related work with respect to information theoretic measures to improve T2I models, and insufficient experimental validation.

**Reviewer Concerns:**

Since there was no rebuttal provided, reviewer's concerns remained unaddressed.

**Reviewer Scores:**

The meta-reviewer believes the reviewers would not have changed their scores due to the absence of any rebuttal.

---

### Decision · Program_Chairs · 2026-01-26

Reject